# Correction of Range-Variant Motion Error and Residual RCM in Sparse Regularization SAR Imaging

**DOI:** 10.3390/s22207927

**Published:** 2022-10-18

**Authors:** Jingyi Zhang, Jiacheng Ni

**Affiliations:** 1Teachering and Research Supporting Center, Air Force Engineering University, Xi’an 710051, China; 2Information and Navigation College, Air Force Engineering University, Xi’an 710077, China

**Keywords:** sparse SAR imaging, motion error correction, range cell migration correction, approximated observation

## Abstract

*L_q_* (0 < *q* ≤ 1) regularization has been confirmed effective when applied to sparse SAR imaging. However, the inaccuracies caused by motion errors in the observation model will lead to various degradations and defocus in the reconstructed image. For high-resolution and light-small SAR systems, the range-variant motion errors will decrease the accuracy of range cell migration correction (RCMC), and residual range cell migration (RCM) will exceed multiple range resolution cells and degrade the image quality substantially. Aiming at this problem, in this paper, a novel azimuth-range decoupled sparse SAR imaging method with coarse-to-fine range-variant motion errors and residual RCM correction method is proposed. First, a one-step motion compensation (MOCO) operator is proposed using the inertial navigation systems (INS)/global positioning systems (GPS) information, which can significantly reduce the residual RCM and improve the reconstruction accuracy. Second, a fine high-order phase-error correction method is performed to correct the range and cross-range-varying phase errors using a joint imaging and phase-error estimation scheme, which will further improve the image focusing quality. Experimental results indicate the effectiveness of the proposed method.

## 1. Introduction

In recent years, a more advanced synthetic aperture radar (SAR) image formation technique called sparse regularization SAR imaging has shown great potential [1,2,3,4,5,6,7]. The accurate reconstruction of sparse regularization SAR imaging is based on the assumption that the observation process of the imaging system is perfectly known [1,5]. However, in practice, the motion error of a radar platform will introduce envelope migrations and phase errors into the echo data, which causes serious degradation to the reconstruction quality. When it comes to high-resolution SAR with small-aircraft platforms, the problem of motion error is even worse. It may yield residual envelop migrations into other range cells, and high-order phase aberrations may exist. In the case of squint mode or high-resolution wide-swath conditions, SAR systems have to continuously increase the synthetic aperture time and expand the instantaneous beam, which causes wide-beam autofocus problems [6]. This also increases the difficulty in motion error correction. In traditional SAR imaging, motion errors can be partly compensated by motion compensation (MOCO) using inertial navigation systems (INS)/global positioning systems (GPS) information and further corrected by data-driven autofocus methods [8,9,10]. However, INS/GPS may not be able to accurately perform MOCO due to the limitations of small-aircraft payloads. Classical autofocus approaches such as phase-gradient autofocus (PGA) [8] and metrics-based autofocus [9] can post-process conventionally reconstructed defocused images and perform well in the case of full aperture or sub-aperture. For wide-beam autofocus problems, Chen et al. [10] proposed a full-aperture autofocus method based on blind RS, which avoided the problems of traditional sub-aperture methods and significantly improved the overall image quality. However, the effect of such methods in the case of incomplete data and sparse aperture needs further research. Therefore, how to estimate and compensate the motion error has become a hot issue in sparse regularization SAR imaging.

Up to now, most existing motion error correction methods for sparse regularization SAR imaging were achieved based on a two-step optimization framework [11], where a joint cost function is iteratively solved to achieve SAR image formation and phase-error estimation simultaneously. Under this framework, Yang et al. [12] proposed a method that can estimate the observation position error instead of phase errors. However, this method used an azimuth-range coupled sparse imaging method, which has high computational complexity. Wu et al. [13] used a sparse Bayesian approach for image reconstruction instead of the norm regularization method and obtained similar results compared with L1 norm regularization. Bu et al. [14] proposed a matrix form regularized smoothed L0 norm (MReSL0) algorithm to solve the joint optimization problem. This method used the MReSL0 algorithm and required high sparsity of the scene. In [15], a total variation (TV) penalty is integrated into the cost function to smooth the target and reduce the noise of the image. Shao et al. [16] compensated the phase error in azimuth direction using under-sampled data and achieve good results. Tian et al. [17] applied the same joint cost function on MIMO radar phase-error correction. 

However, the abovementioned two-step phase-error correction methods can only compensate phase errors that vary along the azimuth direction. This assumption is valid when the range swath is small, i.e., there is no residual range cell migration (RCM). When facing a high-resolution and light-small SAR system, the range-variant motion errors will cause range cell migration correction (RCMC) error, and the residual RCM will degrade the image quality substantially. The above-mentioned phase-error compensating algorithms will fail to deal with a situation where radar echo exists with residual RCM together with cross-range-varying phase errors. To deal with such motion errors in sparse regularization SAR imaging, Kelly et al. [18] completed MOCO and RCMC in range Doppler domain in the first place, and then applied sparse regularization after range compression. The final image was obtained by traditional azimuth matched filtering (MF). The main drawback of [18] is that the sparsity in range direction is not effectively utilized, and the final imaging results do not include the advantages of sparse imaging.

In this paper, we propose a coarse-to-fine correction method of range-variant motion errors and residual RCM for sparse regularization SAR imaging. We first perform a one-step MOCO operator using INS/GPS information as the coarse correction step. We derive the detailed formula expression of residual RCM and range-variant motion errors in the wavenumber domain and design a series of compensation function matrixes to replace 2D interpolation in the conventional one-step MOCO method [19]. Then, in order to further compensate the high-order phase error, a joint cost function of SAR imaging and cross-range-varying phase-error estimation is iteratively solved to achieve finer motion error correction. For sparse regularization SAR imaging, we use an azimuth-range decouple method [6,7] to remove the large size of the range–azimuth coupled observation matrix and change it into multiplications of a series of small-size range–azimuth decoupled matrixes, thus reducing the memory cost and the burden of computing the pseudo-inverse of the observation matrix. Both the proposed MOCO operator and the joint phase-error correction method can be directly added in this decoupled azimuth-range-based sparse SAR imaging method. Note that [20] has applied the two-step phase-error correction method on this azimuth-range decoupled sparse imaging method but still can only deal with the same azimuth phase errors in [12,13,14,15,16,17]. The proposed method has the following advantages:(1)The proposed one-step MOCO operator can significantly reduce the residual RCM, hence improve the reconstruction accuracy of the azimuth-range decoupled sparse SAR imaging method and the focusing quality of SAR images.(2)The proposed method can reduce cross-range-varying phase errors, which will further improve the image focusing quality.

The reminder of this paper is organized as follows. Section 2 introduces the SAR observation model and the azimuth-range decoupled sparse SAR imaging method. Section 3 proposes the coarse-to-fine correction method of range-variant motion errors and residual RCM. Section 4 shows both simulation and raw data results to verify the effectiveness of the proposed approach. Section 5 provides the conclusion.

## 2. Sparse Regularization SAR Imaging Model

### 2.1. SAR Observations Model with Motion Error

The observation geometry of an airborne SAR system is shown in Figure 1, where the solid curve indicates the actual track of the plane and the dashed straight line denotes the nominal path of the platform. The actual and nominal positions of the antenna phase center (APC) are at $\left[X+\Delta x\left(t_{a}\right),\Delta y\left(t_{a}\right),H+\Delta z\left(t_{a}\right)\right]$ and [X,0,H], respectively, where $X=vt_{a}$ is the platform position, $t_{a}$ is the azimuth time and the symbol *o* denotes the scene center. For a target point $P\left(x_{p},y_{p},z_{p}\right)$, the nominal instantaneous range $R_{0}\left(X\right)$ and actual instantaneous range R(X) can be expressed as:(1)$$\begin{array}{ll} R_{0}\left(X\right) & =\sqrt{{\left(X-x_{p}\right)}^{2}+y_{p}^{2}+{\left(H-z_{p}\right)}^{2}}\\ & =\sqrt{{\left(X-x_{p}\right)}^{2}+r^{2}} \end{array}$$
(2)$$R\left(X\right)=\sqrt{{\left(X+\Delta x\left(t_{a}\right)-x_{p}\right)}^{2}+{\left(\Delta y\left(t_{a}\right)-y_{p}\right)}^{2}+{\left(H+\Delta z\left(t_{a}\right)-z_{p}\right)}^{2}}$$
where $r=\sqrt{y_{p}^{2}+{\left(H-z_{p}\right)}^{2}}$ represents the vertical distance from *o* to radar. Under the far-field assumption, the motion errors of the platform can be expressed as:(3)$$\begin{array}{ll} \Delta R\left(X\right) & =R\left(X\right)-R_{0}\left(X\right)\\ & \approx \frac{2X+\Delta x\left(t_{a}\right)-2x_{p}}{r}\cdot \Delta x\left(t_{a}\right)+\sin\theta \cdot \Delta y\left(t_{a}\right)+\cos\theta \cdot \Delta z\left(t_{a}\right) \end{array}$$
where $\sin\theta =\sqrt{1-{\left(H/r\right)}^{2}}$, cosθ=H/r, and θ is the radar incidence angle.

Suppose the SAR system transmits linear-frequency-modulated (LFM) signals, the demodulated echo from point *P* is given by:(4)$$\begin{array}{ll} s\left(\tau ,X\right)= & \sigma_{p}\omega \left(\tau \right)\omega \left(t_{a}\right)\\ & \exp\left[j\pi \gamma {\left(\tau -\frac{2R\left(X\right)}{c}\right)}^{2}-\frac{j4\pi f_{c}R\left(X\right)}{c}\right] \end{array}$$
where τ denotes the range time, and $\sigma_{p}$ denotes the scattering reflectivity coefficient of point *P*. ω(.) represents the envelope function. For simplicity, we omit the specific expression of ω(τ) and $\omega \left(t_{a}\right)$. $T_{p}$ is the pulse duration width, *c* is the velocity of light, $f_{c}$ is the carrier frequency, and γ is the chirp rate. *L* is the synthetic aperture length. We apply range Fourier Transform (FT) with respect to τ and transfer the signal into the range wavenumber domain, it yields:(5)$$\begin{array}{ll} s\left(K_{r},X\right)= & \sigma_{p}\omega \left(K_{r}\right)\omega \left(t_{a}\right)\\ & \exp\left[j\frac{{\left(K_{r}-K_{rc}\right)}^{2}c^{2}}{16\pi \gamma }\right]\exp\left[-jK_{r}R\left(X\right)\right] \end{array}$$
where $K_{r}=K_{rc}+\Delta K_{r}$ denotes the range wavenumber and $K_{rc}=4\pi f_{c}/c$ denotes the range wavenumber center. $\Delta K_{r}\in \left[-2\pi \gamma T_{p}/c,2\pi \gamma T_{p}/c\right]$. After range compression (multiplying $s\left(K_{r},X\right)$ by the negative of first exponential term in (5)), we have:(6)$$\begin{array}{ll} s\left(\tau ,X\right)= & \sigma_{p}\sin c\left[T_{p}\gamma \left(\tau -\frac{2\left(R_{0}\left(X\right)+\Delta R\left(X\right)\right)}{c}\right)\right]\omega \left(t_{a}\right)\\ & \exp\left[-jK_{r}R_{0}\left(X\right)\right]\exp\left[-jK_{r}\Delta R\left(X\right)\right] \end{array}$$

The second exponential term in Equation (6) is induced by the motion errors, including both range-variant motion errors and residual RCM. For high-resolution and light-small SAR systems, the accuracy of the INS/GPS is low, so both errors must be compensated.

### 2.2. Azimuth-Range Decoupled Sparse Regularization SAR Imaging

In regularization-based SAR imaging, (4) can be changed into a more general matrix form as s=Hg+n, where s is the echo data, and H∈ℂNrNa×PQ is the observation matrix that contains the radar phase history and window processing terms. g∈ℂPQ×1 are the discrete vectors of scattering coefficients “σ”. The Nr and Na denote the range samples and azimuth samples, respectively. *PQ* is the number of coefficients after the discretization of the observation scene. n is the system noise. In sparse regularization SAR imaging, g can be exactly recovered by solving a *L_q_* (0 < *q* ≤ 1) norm regularization problem:(7)g^=argming{‖s-Hg‖22+λ‖g‖qq}
where ‖⋅‖q denotes the *L_q_* norm, and λ is the regularization parameter. The main problem of solving Equation (7) is that the observation matrix H owns an intrinsically 2D structure and can cost a huge memory load when the imaging area is large. Solving Equation (7) has a total computation cost of O(Nr⋅Na⋅P⋅Q⋅s) in one iteration, where s is the radar sampling rate. It is known that the radar sampling number in both range direction and azimuth direction is very large, so it is apparent that solving Equation (7) has huge computational costs. This drawback limits the effective applications of sparse regularization SAR imaging.

In order to reduce computational complexity and the required memory cost, Jian et al. [6] presents an azimuth-range decoupled sparse regularization SAR imaging method. It uses a matrix M−1≈H derived from the inverse of MF-based procedures to replace the exact observation matrix H. Since M−1 can be decoupled into a sequence of 1D inverse MF operations, Equation (7) can be changed into a matrix form. In this case, the echo data ***s*** and the unknown reflectivity map g can be maintained as 2D matrices instead of being stacked into vectors. Specifically, let S be the matrix form of radar echo and ***G*** be the matrix form of reflectivity map, Equation (7) can be change into:(8)G˜=minG{‖S−M−1(G)‖22+λ‖G‖q}
where M−1∈ℂNr×Na has a much smaller size than ***H***. Equation (8) can be well solved by an iterative thresholding algorithm (ITA). Owing to the low complexity of MF-based algorithms and the matrix form of regularization optimization, the azimuth-range decoupled imaging algorithms can reduce the computational complexity of the exact observation-based method significantly. However, this method has not considered motion error. In this paper, we put forward an azimuth-range decoupled sparse SAR imaging method that can deal with range-variant motion errors and residual RCM.

## 3. Proposed Motion Error Correction Method

### 3.1. One-Step MOCO Operator

Using the INS/GPS information, we put forward a one-step MOCO operator and add it in $M^{-1}$ to process coarse motion error correction. We first derive the detailed formula expression of range-variant motion error and residual RCM in the wavenumber domain. Then, we use these formula expressions to design compensation functions for our one-step MOCO operator. For simplicity, we only consider the exponential terms of received signal in the following deduction.

Applying range FT to Equation (6), omit the envelope terms, it gives:(9)$$\begin{array}{ll} s\left(\Delta K_{r},X\right) & =\int s\left(\tau ,X\right)\exp\left(-j\frac{\tau c}{2}\Delta K_{r}\right)d\frac{\tau c}{2}\\ & =\exp\left(-jK_{r}\left[R_{0}(X)+\Delta R(X)\right]\right) \end{array}$$

We then apply azimuth FT with respect to *X*, it gives:(10)$$s\left(\Delta K_{r},K_{x}\right)=\int s\left(\Delta K_{r},X\right)\exp\left[-jK_{x}X\right]dX$$

To deduce the detailed analytic expression of 2D point target reference spectra, here, we use the principle of stationary phase (POSP). Equating the first derivative of the phase function with respect to *X* in Equation (10) to X=0 gives:(11)$$K_{r}\frac{\left(X-x_{p}\right)}{R_{0}\left(X\right)}+K_{r}\frac{\partial \Delta R\left(X\right)}{\partial X}+K_{x}=0$$

Using the approximation to ignore the impact of ΔR(X) to the stationary phase point [19] (this approximation is usually valid when the synthetic aperture is not too long), a simplified solution of stationary phase point can be expressed as:(12)$$X^{*}\approx \frac{K_{x}}{\sqrt{K_{r}^{2}-K_{x}^{2}}}r+x_{p}$$

Then, by substituting $X^{*}$ for X in Equation (10), the expression of signal in two-dimensional wavenumber domain is given as:(13)$$\begin{array}{ll} s\left(\Delta K_{r},K_{x}\right)= & \exp\left[-j\left(r\cdot \sqrt{K_{r}^{2}-K_{x}^{2}}+K_{x}\cdot x_{p}\right)\right]\\ & \exp\left[-jK_{r}\Delta R\left(X^{*}\right)\right] \end{array}$$
where $K_{x}$ denotes the azimuth wavenumber and $X^{*}$ denotes the stationary phase point. Then, we expand the first phase term of Equation (12) by Taylor expansion with respect to $\Delta K_{r}=K_{r}-K_{rc}$ around $\Delta K_{r}=0$, Equation (13) can be expressed as:(14)$$\begin{array}{ll} s\left(\Delta K_{r},K_{x}\right)= & \exp\left[-j\left(r\cdot \sqrt{K_{rc}^{2}-K_{x}^{2}}+K_{x}\cdot x_{p}\right)\right]\exp\left[-jK_{r}\Delta R\left(X^{*}\right)\right]\\ & \exp\left[-j\frac{K_{rc}\cdot r}{\sqrt{K_{rc}^{2}-K_{x}^{2}}}\Delta K_{r}\right]\exp\left[j\frac{K_{x}^{2}\cdot r}{2{\left(K_{rc}^{2}-K_{x}^{2}\right)}^{3/2}}\Delta K_{r}^{2}\right] \end{array}$$
where the third exponential term in Equation (14) is the RCM term and the fourth term is the second-order range–azimuth coupling phase term. Thus, we can obtain the RCMC function and the second-order range compression function as: (15)$$H_{RCMC}=\exp\left[j\left(\frac{K_{rc}\cdot r}{\sqrt{K_{rc}^{2}-K_{x}^{2}}}-r\right)\Delta K_{r}\right]$$
(16)$$H_{SRC}=\exp\left[-j\frac{K_{x}^{2}\cdot r}{2{\left(K_{rc}^{2}-K_{x}^{2}\right)}^{3/2}}\Delta K_{r}^{2}\right]$$

In Equation (14), the second exponential term represents the residual motion error phase term. In conventional two-step MOCO, this term is compensated after RCMC, which will cause a serious RCMC error. Here, we use POSP to deduce the detail analytic expression of the relationship between residual motion error and RCMC error. Expand this residual motion error term $\exp\left[-jK_{r}\Delta R\left(X^{*}\right)\right]$ by Taylor expansion with respect to $\Delta K_{r}$ around $\Delta K_{r}=0$, we can obtain:(17)$$\Delta R\left(X^{*}\right)=e_{1}+e_{2}\Delta K_{r}+e_{3}\Delta K_{r}^{2}+\ldots$$

In Equation (17), the high-order terms have little effect on RCMC performance, so we only keep the constant term and the first-order term, where the coefficients are expressed as:(18)$$\begin{array}{ll} e_{1} & =\Delta R\left(X^{*}\right)|{}_{\Delta K_{r}=0}\\ e_{2} & =\frac{\partial \Delta R\left(X^{*}\right)}{\partial \Delta K_{r}}|{}_{\Delta K_{r}=0}=\frac{\partial X^{*}}{\partial \Delta K_{r}}\cdot \frac{\partial \Delta R\left(X^{*}\right)}{\partial X^{*}}\\ & =\frac{K_{x}\left(K_{rc}+\Delta K_{r}\right)\cdot r}{2{\left[{\left(K_{rc}+\Delta K_{r}\right)}^{2}-K_{x}^{2}\right]}^{3/2}}\cdot \frac{\partial \Delta R\left(X^{*}\right)}{\partial X^{*}} \end{array}$$

Applying azimuth IFT to Equation (14) to transfer the signal to the range wavenumber domain, we have:(19)$$\begin{array}{l} s\left(\Delta K_{r},X\right)=\int s\left(\Delta K_{r},K_{x}\right)\cdot \exp\left[jK_{x}X\right]dK_{x}\\ =\exp\left[-jK_{rc}R_{0}\left(X\right)\right]\exp\left[\Delta R\left(K_{x}^{*}\right)\cdot \left(K_{rc}+\Delta K_{r}\right)\right] \end{array}$$
where $K_{x}^{*}$ represents the stationary phase point. Since the RCM term and SRC term in Equation (14) will be compensated, we ignore these two phase terms when calculating $K_{x}^{*}$. We also use the approximation to ignore the impact of $\Delta R\left(K_{x}^{*}\right)$ to the stationary phase point, thus $K_{x}^{*}$ is given by:(20)$$K_{x}^{*}\approx -K_{rc}\frac{X-x_{p}}{\sqrt{r^{2}+{\left(X-x_{p}\right)}^{2}}}=-K_{rc}\frac{X-x_{p}}{R_{0}\left(X\right)}$$

Using the expression of $K_{x}^{*}$ in Equation (20) to replace $K_{x}$ in Equation (18), the residual motion error in azimuth time and range wavenumber domain can be expressed as:(21)$$\Delta R\left(X\right)=\Delta R\left(r\right)+\frac{\left(X-x_{p}\right)\left[r^{2}+{\left(X-x_{p}\right)}^{2}\right]}{r^{2}K_{rc}}\Delta K_{r}$$
(22)$$\begin{array}{ll} \Delta R\left(r\right) & =\frac{X-x_{p}}{r}\Delta x+\Delta y\cdot \sin\beta +\Delta z\cdot \cos\beta \\ & =\frac{X-x_{p}}{r}\Delta x+\frac{\sqrt{r^{2}-H^{2}}}{r}\Delta y+\frac{H}{r}\Delta z \end{array}$$
where ΔR(r) represents the azimuth-invariant motion error in slant range. Δx, Δy, and Δz represent the along track error, cross-track error, and height error, respectively. β denotes the target look angle. The second term in Equation (21) is the residual range term generated by the first-order term of Taylor expansion.

After we obtain ΔR(X), we can replace $\Delta R\left(K_{x}^{*}\right)$ with ΔR(X), then the expression of signal in azimuth time and range wavenumber can be obtained as:(23)$$\begin{array}{l} s\left(\Delta K_{r},X\right)=\exp\left[-jK_{rc}R_{0}\left(X\right)\right]\exp\left[-j\left(K_{rc}\Delta R\left(r\right)+\left(r\left(x\right)+\Delta R\left(r\right)\right)\Delta K_{r}+r\left(x\right)\Delta K_{r}^{2}\right)\right]\\ r\left(x\right)=\left(X-x_{p}\right)\left[r^{2}+{\left(X-x_{p}\right)}^{2}\right]/r^{2} \end{array}$$

Equation (23) shows the relationship between the residual motion error and the RCMC error, where the second exponential term of Equation (23) denotes the RCMC error that relates to the residual motion error. It is noted that the second exponential term of Equation (23) is expressed as a series order of $\Delta K_{r}$, and the second-order term has little impact on the focusing performance. So, we ignore the second-order term of $\Delta K_{r}$ and simplify Equation (23) as:(24)$$s\left(\Delta K_{r},X\right)=\exp\left[-jK_{rc}R_{0}\left(X\right)\right]\exp\left[-jK_{rc}\Delta R\left(r\right)\right]\exp\left(-j\left[\Delta R\left(r\right)+r\left(X\right)\right]\Delta K_{r}\right)$$

In Equation (24), the second and the third term denote the phase component and the envelope component of RCMC error. It is known that in the one-step MOCO method, the residual range-variant motion error is compensated before RCMC, so both the second and the third term in Equation (24) should be compensated before RCMC. The envelope compensation is processed in range wavenumber domain by multiplying:(25)$$H_{MOCO\_E}\left(K_{r},X\right)=\exp\left[j\left[\Delta R\left(r\right)+r\left(X\right)\right]\Delta K_{r}\right]$$

Then, the phase compensation is processed in range time domain by multiplying:(26)$$H_{MOCO\_P}\left(\tau ,X\right)=\exp\left[-jK_{rc}\Delta R\left(r\right)\right]$$

From Equations (17)–(25), we derived the detailed analytical expression of residual RCM which is related to the range-variant motion error, and designed a series of compensation functions to deal with range-variant motion error and residual RCM. We now use these compensation functions to construct the one-step MOCO operator in $M^{-1}$. We first derive matrix ***M***. Define range FT and IFT operators as $F_{r}$ and $F_{r}^{H}$, respectively. Define two-dimensional FT and IFT operators as $F_{2D}$ and $F_{2D}^{H}$, respectively. Then, according to Figure 2, matrix ***M*** can be written as:(27)$$M(S)=\{H_{RCMC}\circ H_{SRC}\circ \{F_{2D}\{H_{MOCO\_P}\circ <[H_{MOCO\_E}\circ H_{r}\circ (SF_{r})]F_{r}^{H}>\}\}\}F_{2D}^{H}$$
where $H_{r}$ represents the range compression operator, and “∘” is the Hadamard multiplication. Then, we take the inverse of every procedure in Equation (27) to derive $M^{-1}$. It is known that the inverse of FT is IFT, and the inverse of phase multiplication ***H*** is the multiplication of conjugate phase $H^{*}$, so $M^{-1}$ can be expressed as:(28)$$M^{-1}(G)=\left\{H_{MOCO\_E}^{*}\circ H_{r}^{*}\circ \left\{\left\{H_{MOCO\_P}^{*}\circ \langle \left[H_{RCMC}^{*}\circ H_{SRC}^{*}\circ (SF_{2D})\right]F_{2D}^{H}\rangle \right\}F_{r}\right\}\right\}F_{r}^{H}$$

We call matrix ***M*** and $M^{-1}$ in Equations (27) and (28) one-step MOCO operators. If the trajectory information of the platform is accurately recorded by the INS/GPS, the proposed one-step MOCO operator can compensate most of the range-variant motion error and make the range error constrained within one range cell. However, the requirement of accurate radar phase history is usually beyond the capability of common INS/GPS systems. Hence, after coarse MOCO, we still need to correct the phase errors caused by INS/GPS uncertainty. In the next section, we present a joint imaging and cross-range-varying phase-error correction method.

### 3.2. Joint Imaging and Phase-Error Correction

In this section, we take range and cross-range-varying phase errors into account, because the residual motion errors and residual RCM will affect both range and cross-range directions. Assume that all samples of phase history data contain different phase errors. Define $E_{2D}$ as the 2D phase-error matrix. In this case, the azimuth-range decoupled sparse SAR imaging model with range and cross-range-varying phase errors becomes:(29)$$\tilde{G}=\underset{G}{\min}\left\{{\left\|S-E_{2D}\circ M^{-1}(G)\right\|}_{2}^{2}+\lambda {\left\|G\right\|}_{q}\right\}$$
(30)$$E_{2D}=\left[\begin{array}{cccc} \exp\left(je_{1,1}\right) & \exp\left(je_{1,2}\right) & \cdots & \exp\left(je_{1,N_{a}}\right)\\ \exp\left(je_{2,1}\right) & \cdots & & \\ \vdots & & \cdots & \\ \exp\left(je_{N_{r},1}\right) & & & \exp\left(je_{N_{a}N_{r}}\right) \end{array}\right]$$
where $e_{m,n}$ represents the parameter of phase errors at *m*-th range samples and *n*-th azimuth samples. The goal of phase-error correction is to find the best estimate of $E_{2D}$ and remove it from the phase history data. Following the proposed one-step MOCO operator, the problem of joint imaging and phase-error correction can be solved by minimizing the following cost function:(31)$$J\left(E_{2D},G\right)=\underset{G,E_{2D}}{\min}\left\{{\left\|S-E_{2D}\circ M^{-1}(G)\right\|}_{2}^{2}+\lambda {\left\|G\right\|}_{q}\right\}$$

The given cost function is iteratively minimized with respect to ***G*** and $E_{2D}$ by solving a two-step optimization problem. In step 1, ***G*** is reconstructed by the given phase error using the sparse imaging method. The optimization problem can be expressed as:(32)$$G^{\left(i+1\right)}=\underset{G}{\min}\left\{{\left\|S-E_{2D}^{{}^{\left(i\right)}}\circ M^{-1}(G)\right\|}_{2}^{2}+\lambda {\left\|G\right\|}_{q}\right\}$$
where $G^{\left(i\right)}$ and $E_{2D}^{{}^{\left(i\right)}}$ represent the imaging result and the estimated 2D phase error in the *i*-th iteration, respectively. In step 2, the phase-error matrix $E_{2D}$ is estimated with the fixed ***G***. Since $\lambda {\left\|G\right\|}_{q}$ in this step is a constant, the optimization problem is expressed as:(33)$$E_{2D}^{\left(i+1\right)}=\underset{E_{2D}}{\min}{\left\|S-E_{2D}\circ M^{-1}(G^{\left(i+1\right)})\right\|}_{2}^{2}$$

According to Equation (30), Equation (33) can be separated into a set of optimization problems as follows:(34)$$e_{m,n}^{\left(i+1\right)}=\underset{e}{\mathrm{argmin}}{\left\|S_{\left(m,n\right)}-\exp\left(je_{m,n}\right){\left[M^{-1}(G^{\left(i+1\right)})\right]}_{\left(m,n\right)}\right\|}_{2}^{2}$$
where ${\left[\;\right]}_{\left(m,n\right)}$ denotes the *m*-th row and *n*-th column of the matrix. The solution of the optimization problem in (34) is as follows:(35)$$e_{m,n}^{\left(i+1\right)}=\mathrm{angle}\left(S_{\left(m,n\right)}\cdot {\left[M^{-1}(G^{\left(i+1\right)})\right]}_{\left(m,n\right)}\right)$$

After obtaining E2D(i), let *i = i* + 1 and turn back to step 1 in Equation (32). The iteration stops when ‖G(i+1)−G(i)‖2/‖G(i)‖2 is less than a determined threshold *T*. The detailed procedure of joint imaging and phase-error correction is summarized in Algorithm 1. We use the same parameter selecting method of λ and μ in [6]. The overall flow chart of the entire imaging method is shown in Figure 3.
**Algorithm 1.** The proposed joint imaging and phase-error correction method.**Input: *S*, *M*,**M−1**Initialization:**G(0), E2D(0)=I, λ,μ, iterations Itotal, *q* = 1, threshold *T*.**For**
*i* = 1: *I*_total_, **do** **Step 1**: Imaging part, given E2D(i), calculate residue: R^(i)=S−E2D(i)∘[M−1(G(i))]Residue of one-stepMOCO: ΔG(i)=M(R(i))Thresholding: G(i+1)=Th1,λμ(G(i)+μM(R(i)))Th1,λμ(x)={sgn(x)(|x|−λμ)|x|≥λμ0otherwise **If**
‖G(i+1)−G(i)‖2/‖G(i)‖2<T
**then break** **else** **Step 2**: Phase-error estimation part, given ***G***^(*i*+1)^, calculate Equation (35), obtain em,ni+1;Update E2D(i+1)=minE2D‖S−E2D∘M−1(G(i+1))‖22**end if****end for****Output**: well-focused image ***G***

### 3.3. Analysis of Computational Complexity and Memory Cost

The analysis of the computational complexity of the proposed sparse regularization SAR imaging method is provided as follows. Let $N_{r}$ and $N_{a}$ denote the range samples and azimuth samples, and define $I_{\mathrm{total}}$ as the total number of iterations. For the imaging step, one iteration includes the calculation of one-step MOCO operator, the calculation of residue, and the thresholding operation. The calculations of one-step MOCO operator and residue have the same computational complexity as $O\left[N_{r}N_{a}\log_{2}\left(N_{r}N_{a}\right)\right]$ [6]. The complexity of thresholding is order $O\left(N_{r}N_{a}\right)$. Assume that the number of iterations in this step is $I_{1}$, and the total computational complexity of the imaging step is $O\left[I_{1}N_{r}N_{a}\log_{2}\left(N_{r}N_{a}\right)\right]$. For the phase-error estimation step, as can be seen in (34), the computational complexity is order $O\left(N_{r}N_{a}\right)$. To sum up, the total computational complexity of the proposed method is at order $O\left[I_{\mathrm{total}}I_{1}N_{r}N_{a}\log_{2}\left(N_{r}N_{a}\right)\right]$.

Regarding the memory cost of the proposed method. Since the range and cross-range-varying phase-error estimation step can be separated into a set of independent optimization problems, the biggest matrix will appear in the imaging step. The size of the observation matrix in the one-step MOCO operator is $N_{r}N_{a}$. The size of the exact observation matrix is $N_{r}N_{a}\times PQ$, where PQ is the number of discretized coefficients of the imaging scene. Therefore, the proposed method can reduce the memory cost significantly.

## 4. Applications Results

### 4.1. Point Target Simulations

We first conduct point target simulations to demonstrate the effectiveness of the proposed SAR imaging method. The parameters of the simulated SAR system and two point target coordinates are shown in Table 1. The echo data were generated in time domain by exact slant range mixed with motion errors extracted from a real−measured airborne INS. The detailed motion errors are shown in Figure 4a.

We add complex white Gaussian noise to the echo data so that the input signal-to-noise ratio (SNR) is 15 dB. It is noted that points 1 and 2 are at the edge of the scene, i.e., the range-variant motion error and residual RCM are obvious. The distance between points 1 and 2 is set close to test the sidelobe suppression ability of the proposed method. Figure 4b shows the imaging result using the chirp scaling algorithm (CSA) without MOCO, where motion errors cause serious defocus in both point targets.

Figure 5 presents the RCMC result of the proposed one-step MOCO method. It can be seen in Figure 5b that the residual RCM after the conventional RCMC process exceeds multiple range bins, which will certainly cause a big reconstruction error in sparse regularization SAR imaging. It is shown in Figure 5c that all the residual RCM of points 1 and 2 are well-removed after using the proposed one-step MOCO method, which also indicates that the majority of the motion errors are removed.

Figure 6 presents the imaging results using different imaging methods. Figure 6a shows the imaging result using CSA with PGA applied. It can be seen that the focusing quality using PGA is not good, and points 1 and 2 are still attached. This is because conventional autofocus algorithms cannot completely compensate for the range-variant motion errors. Figure 6b shows the imaging result using CSA with the proposed one-step MOCO method, where both point targets are well-focused. This proves that the proposed one-step MOCO method can well compensate both range-variant motion error and residual RCM, however, there exists a high level of sidelobes in Figure 6b. Figure 6c shows the imaging result using a joint azimuth-range decoupled sparse imaging algorithm and an azimuth-varying phase-error correction method (method in [20]). The sparsity parameter is set as *K* = 6, the maximum iteration step is set as *I* = 40, and the determined threshold is $T=1\times e^{-6}$. It can be seen that the method in [20] fails to reconstruct points 1 and 2 in the right coordinate. This is because [20] only considered platform velocity error and can only compensate azimuth-varying phase errors. Figure 6d shows the imaging result of the proposed method, where both point targets can be well reconstructed in the right coordinates with no sidelobes.

We then present the quantitative evaluation of the reconstruction results for various input SNRs, as shown in Figure 6. Since the method in [19] fails to reconstruct points 1 and 2, we choose CSA with the proposed one-step MOCO method (written as one-step MOCO method in Figure 7 as well as PGA as comparison methods). We conducted the experiments 20 times to obtain the points on the curves. Figure 7a shows the mean square error (MSE) between the reconstruction image without motion error and the imaging results from different imaging methods. Figure 7b shows the comparison results of target-to-background ratio (TBR), which is used to determine the accentuation of the target pixels with respect to the background. From Figure 7, it is clearly noticeable that the proposed method performs better than the PGA and one-step MOCO method. It also can be seen that the proposed method can provide a very accurate reconstruction of the original scene when SNR is low, which demonstrate the robustness of the proposed method.

### 4.2. Real Data Experiments

In this experiment, we use real measure data in spotlight mode to demonstrate the effectiveness of the proposed method. We use PGA and sparse imaging method in [19] as comparison method.

First, we perform an experiment to validate the performance of the proposed method when the platform trajectory is accurately recorded by the INS. The echo data are obtained from a C-band airborne SAR system, where the system PRF is 2000 Hz, and the signal bandwidth is 200 MHz. The azimuth resolution is about 0.15 m. The full-sized spotlight SAR image is too big and lacks sparsity for sparse regularization SAR imaging, so we choose one typical sparse scene which covers only a sub-swath in azimuth to process imaging. This scene (named as scene A) has an obvious point-like structure and is sparse enough for the sparse SAR imaging method. We discretize scene A into 1024 × 1024 and set the same sparsity parameter for both the method in [19] and the proposed method. The maximum iteration is set as 100. For the PGA method, the initial window width is set to be 512 azimuth samples.

Figure 8 shows the final imaging results of scene A using the PGA method in [19] and the proposed method, respectively. We also put the zoomed-in images containing a dominant point marked with a red square below the imaging result. It can be clearly seen that the proposed method obtained the best image focusing quality. It is noted that in Figure 8b, there is a shift of point target location in the red square window, i.e., the dominate point is not at the same coordinate values compared with Figure 8a,b. Since our main purpose in Figure 8 is to compare the image focusing quality, we adjust the point target to the image center. This phenomenon shows that the motion errors in scene A not only generate azimuth-varying phase errors but also generate other envelope terms and phase errors.

Furthermore, we use another echo dataset to test the effectiveness of the proposed method, where the exact motion information cannot be exploited for MOCO. The data are collected by an X-band airborne SAR system; the platform speed is about 200 m/s and the radar height is about 11,000 m. We name this scene as scene B and discretize it into 1024 × 1024 when using the sparse imaging method. Figure 9 shows the final imaging results. It can be seen that the imaging result using PGA is defocused, and there exists severe deformation in the reconstructed image using the method in [20]. This is because both comparison methods cannot compensate large range-variant motion errors. Figure 9c shows the reconstruction result using the proposed method. It can be seen that the whole image is well-focused. To give a close insight of the reconstruction result, we also show the zoomed-in images containing man-made buildings marked with a red rectangle. It can be seen that the proposed method can give a much clearer visual effect than the competing methods, where the edge of the building is clear and sharp.

We further use image entropy to quantitatively evaluate the performance of the proposed method. All the experiments were conducted on a workstation (Intel Xeon 3.2 GHz, 4 GB RAM) with MATLAB R2016b. Table 2 shows the entropy values of the reconstructed images by different methods. It can be seen that the proposed method has the lowest entropy, which proved the superiority of the proposed method. Table 2 also lists the corresponding running times of the three imaging methods. It can be seen that the proposed method is a little slower than the method in [20] but with nearly equal computation complexity. Compared with the method in [20], the proposed method adds several times of FFT/IFFT operations in the one-step MOCO operator and Na times of phase-error estimation in the phase-error correction step. Since no big matrix multiplications or conjugations are added in the abovementioned steps, the computational cost of the proposed method did not increase significantly.

From the above experiments, it can be seen that the proposed algorithm is reliable to produce good image focusing quality at an affordable computational cost.

## 5. Conclusions

In this paper, we proposed a coarse-to-fine correction method of range-variant motion errors and residual RCM for sparse regularization SAR imaging. First, the precise formula expression between the residual range-variant motion errors and the RCMC error is derived in the wavenumber domain. By doing so, we obtained the corresponding residual RCM and motion error compensation functions using INS/GPS information. We use these compensation functions to construct a one-step MOCO operator. Then, a joint imaging and azimuth-range-varying phase-error correction method is used to perform the fine phase-error compensation. Experimental results of different experimental data objects, real point targets, and real urban cases demonstrate the superiority of the proposed method over comparison methods in terms of image focusing quality. The proposed method has the following advantages.

(1)The proposed one-step MOCO operator can significantly reduce the residual RCM, hence improve the reconstruction accuracy of the azimuth-range decoupled sparse SAR imaging method and the focusing quality of SAR images.(2)The proposed method can reduce cross-range-varying phase errors, which will further improve image focusing quality.

Future research will focus on how to obtain good focusing results when facing wide-beam autofocus problems and how to improve the focusing quality when the INS/GPS information is inaccurate. One possible way is to subsequently model the wide-beam SAR autofocus problem as a multiple-dimensional optimization problem and estimate different unknown parameters to correct the spatial variation of the quadratic, cubic, and quartic phase errors. The accurate estimation of such optimization problems can finally realize wide-beam SAR autofocus processing.

## Figures and Tables

**Figure 1 sensors-22-07927-f001:**
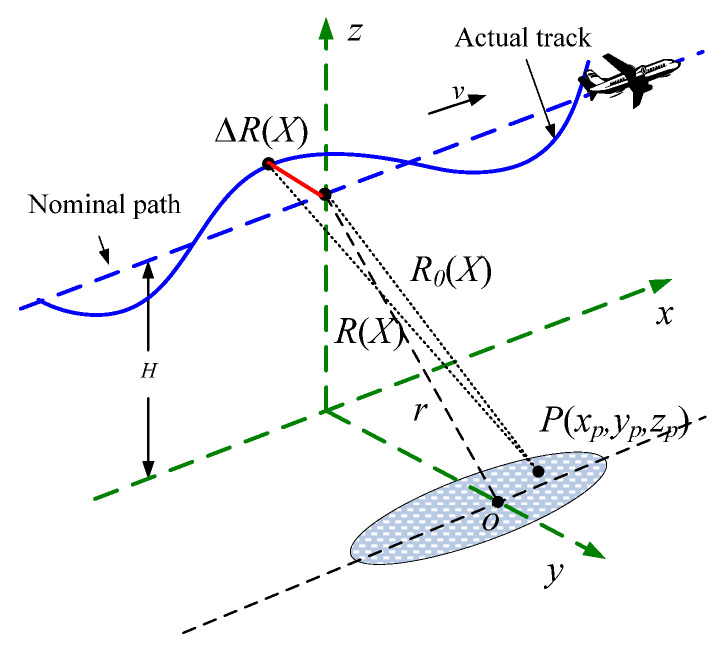
SAR imaging geometry with motion error.

**Figure 2 sensors-22-07927-f002:**
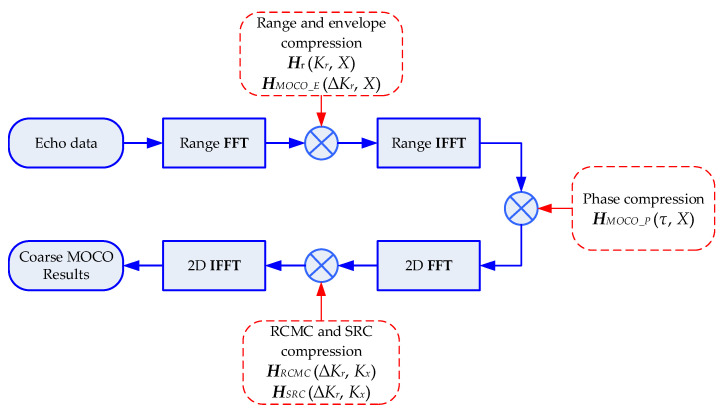
Flowchart of the proposed one-step MOCO operator.

**Figure 3 sensors-22-07927-f003:**
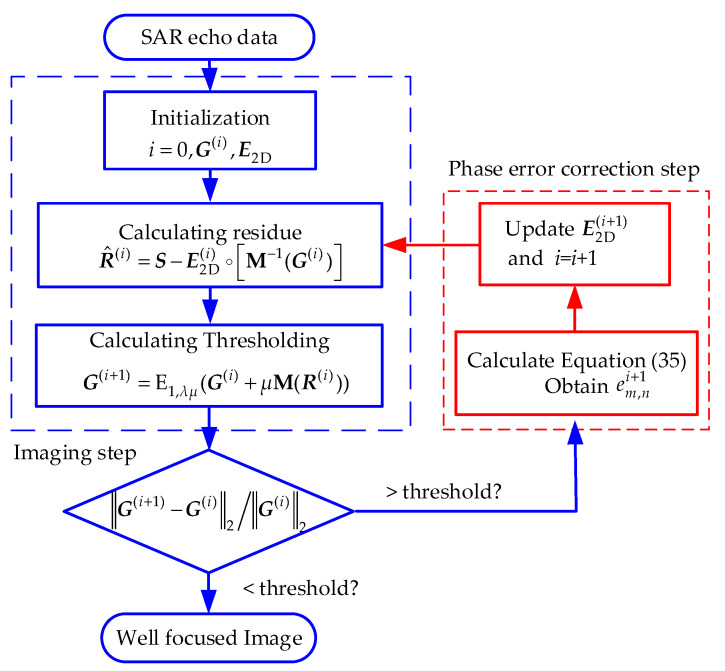
The overall flow chart of the entire sparse SAR imaging and motion error correction method.

**Figure 4 sensors-22-07927-f004:**
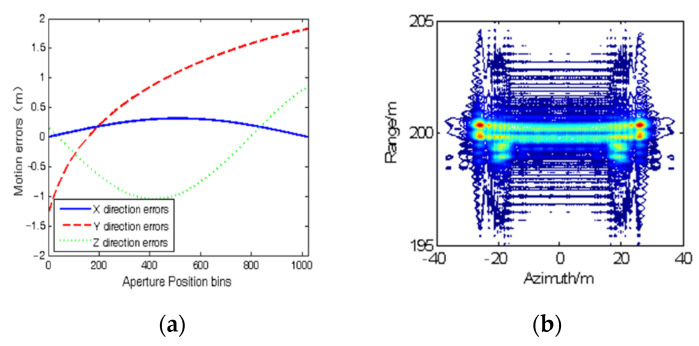
(**a**) Measured motion errors in different directions; (**b**) imaging results of points 1 and 2 using CSA without MOCO.

**Figure 5 sensors-22-07927-f005:**
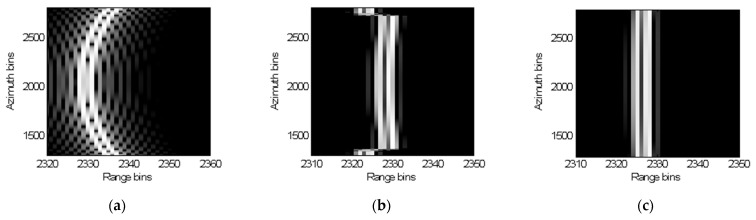
Comparison of range profiles of points 1 and 2. (**a**) Range profile of points 1 and 2 before RCMC; (**b**) RCMC results without using one-step MOCO method; (**c**) RCMC results after one-step MOCO method.

**Figure 6 sensors-22-07927-f006:**
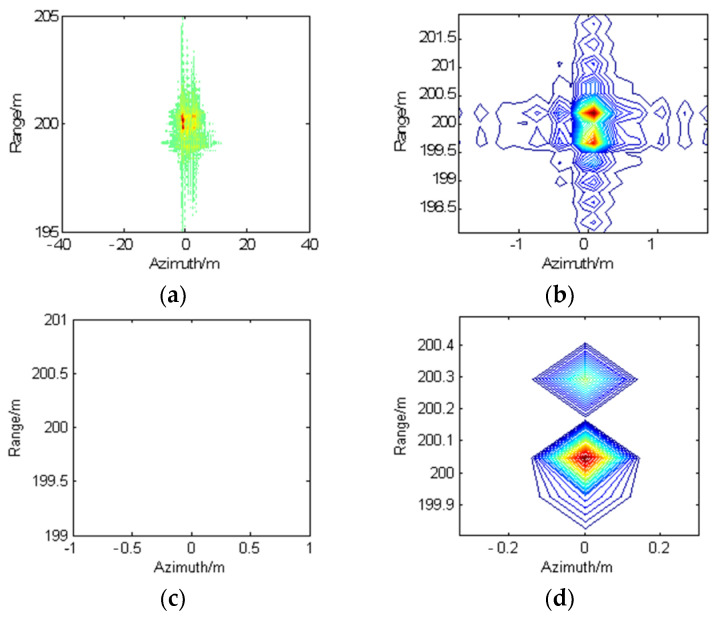
Point targets imaging results, (**a**) CSA with PGA applied; (**b**) CSA with proposed one-step MOCO method; (**c**) imaging method in [20]; (**d**): proposed imaging method.

**Figure 7 sensors-22-07927-f007:**
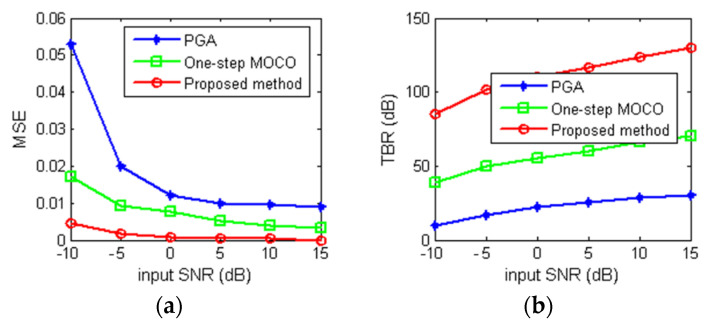
Quantitative evaluation of reconstruction results under various input SNRs. (**a**) MSE results; (**b**) TBR results.

**Figure 8 sensors-22-07927-f008:**
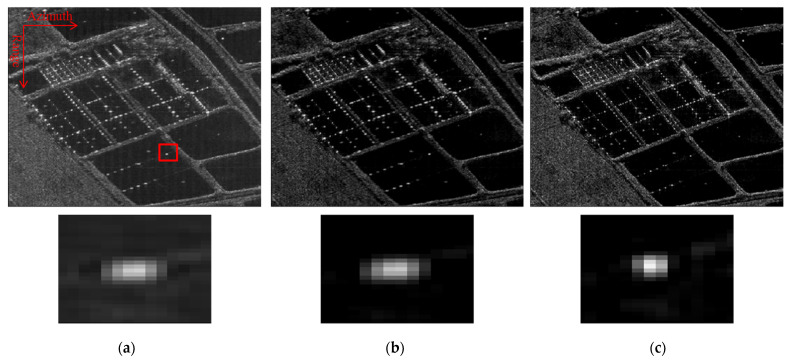
Imaging results of scene A using different methods. (**a**) PGA; (**b**) method in [20]; (**c**) proposed method.

**Figure 9 sensors-22-07927-f009:**
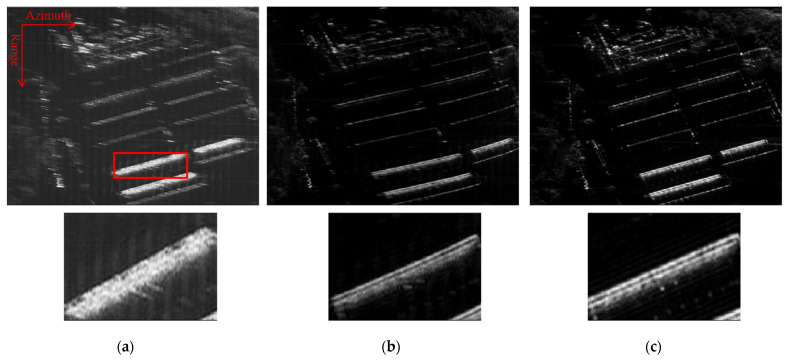
Imaging results of scene B using different methods. (**a**) PGA; (**b**) method in [20]; (**c**) proposed method.

**Table 1 sensors-22-07927-t001:** Parameters of simulated SAR system.

Parameters	Value
Carrier frequency	10 GHz
Bandwidth	500 MHz
Radar velocity	110 m/s
Pulse Reputation Frequency	672
Center slant range	5000 m
Scene width	420 m
Range resolution	0.3 m
Azimuth resolution	0.3 m
Point 1 coordinate	(0, 200)
Point 2 coordinate	(0, 200.3)

**Table 2 sensors-22-07927-t002:** Image entropy and running times of different imaging methods.

Method	Image Entropy		Running Time (s)	
	Scene A	Scene B	Scene A	Scene B
PGA	10.8083	11.4233	1104.5	1766.5
Method in [20]	10.6433	11.2807	635.4	638.8
Proposed method	10.2944	11.0133	675.1	680.3

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
