# Peer review of "Correction of Range-Variant Motion Error and Residual RCM in Sparse Regularization SAR Imaging"

_sensors, 2022, doi:10.3390/s22207927_

Round 1

Reviewer 1 Report

Dear respected Editor and authors

This paper proposes a coarse-to-fine correction method of range variant motion errors and residual RCM for sparse regularization SAR imaging. The authors first perform a one- step MOCO operator using INS/GPS information as the coarse correction step. they derived the detailed formula expression of residual RCM and range variant motion errors in wave number domain, and designed a series of compensation function matrixes to replace 2D interpolation in conventional one step MOCO method. Then, in order to further compensate the high order phase error, a joint cost function of SAR imaging and cross-range-varying phase error estimation was iteratively solved to achieve finer motion error correction. For sparse regularization SAR imaging, they use an azimuth-range decouple method to remove the large size of range-azimuth coupled observation matrix, and change it into multiplications of a series of small size range-azimuth decoupled matrixes, thus reducing the memory cost and the burden of computing the pseudo-inverse of observation matrix. Both the proposed MOCO operator and the joint phase error correction method can be directly added in this azimuth-range decouple based sparse SAR imaging method. The two-step phase error correction method has been applied on the azimuth-range decoupled sparse imaging method, but still can only deal with the same azimuth phase errors.

References are relatively updated, however I believe it is short listed and require to add more reference resources that you already used to derive 35 equations.

I have the following suggestions to improve the paper;

1.      Page 1-line 8: change “effectively” to effective

2.      Page 1-line 14: change “exceeds” to exceed

3.      Page 1-line 31: change “seriously degradation of reconstruction quality” to “serious degradation to reconstruction quality”

4.      Page 2-lines 46-52: change “[12]” to yang et al. [12] . Please do the same for references [13], [14], [16], and [17]

5.      Page 2-line 59: change “when” to where

6.      Page 2-line: change “[18]” as describes in # 4

7.      Page 3-line 99-105: Please double check the accuracy of presentation and symbols in all equations

8.      Page 4-line 120: change “σ . Nr and Na …” to “σ”. The Nr and Na …

9.      Page 4-line 125: change “solving (7)” to solving Equation (7) and do the same for the similar cases.

10.   Line 128: change [6] as described in # 4 and do the same for the similar cases.

11.   Line 139: change “computational complexity” to computation complexity

12.   Line 140-412: please consider rephrasing “However, this method hasn’t considered motion error, let alone range variant motion errors and residual RCM. This motivates us to present our work.”

13.   Line 146: change “detail formula” to detailed formula

14.   Line 153: change “detail analytic expression” to detailed analytical expression.

15.   Line 155: change “it gives: to “gives:”.

16.   Line 155: change “cause serious RCMC error” to cause a serious RCMC error.

17.   Line 177-180: I am not sure if these multiple ignoring steps would influence the model accuracy ???? !!!!!!!

18.   Line 200-201: change “detail analytic expression of residual RCM which 200 related to the range variant motion error,” to “detailed analytical expression of residual RCM which is related to the range variant motion error,”.

19.   Line 266: change “Next, we discuss the memory cost of the proposed method.” to” Regarding the memory cost of the proposed method,”.

20.   Line 277: Omit “which”

21.   Line 301: change “point 1 & 2 is inseparable.” to “points 1 & 2 are still attached.”

22.   Line 351: change “exists” to “is”

23.   Line 359: change “which” to “with”.

24.   Line 370: Table 2 is presented before mentioning it in the text. Please move it down this paragraph in linr 371

25.   Line 375: please start the results section with this sentence in line 375 “All the experiments were conducted on a 375-work station (Intel Xeon 3.2 GHz, 4 GB RAM) with MATLAB R2016b.

26.   Line 385: I believe “Discussion” would be “Conclusion”.

27.   In conclusion section please refer to the three different experimental data object, real point target and real urban case.

I wish this helps improving this work

Good luck

Author Response

The authors are grateful for the reviewer’s valuable comments which have helped us to understand the paper in depth and improve the original manuscript. We have studied the comments carefully and made some corrections which we hope meet with approval. All the revised contents are highlighted in red in the new manuscript. We also highlight the responses in blue colors. The detailed response is in the attachment file.

Reviewer 2 Report

1. Please check the case of the Keywords.

2. In the second paragraph of Section 1, I suggest you add the authors name of literature [12-17].

3. Please explain the abbreviation RCMC and RCM when it occurs first in Section 1.

4. In Line 65 and 81, the format of have is wrong.

5. Please give the full name of INS/GPS in Section 1.

6. The tense of word proposed in Line 86 is false.

7. The reference format of the picture serial number in the text is not standard and unified, please check.

8. Please present the simplest format of the expression (4-5).

9. What meaning is the rectangle in Line 118 and 119.

10. The symbol of matrix may be not italic, please check the requirements of Sensors.

11. Please check and adjust the structure of “if…else…end” in Algorithm 1.

12. The style of Table 1 and 2 differ from the journal requirements, please amend.

13. Why Figure 5(c) is null, please explain. And in Figure 5(b) and (c), why is the estimation accuracy of the Point 1 worse than the Point 2.

14. The title of “4.1. Real data experiments should be “4.2. Real Data Experiments”.

15. It is suggested that revise Section 5 to conclusion. And the discussion should further discuss the advantages and disadvantages of the proposed method.

16. Therare some extra words in Line 402 to be deleted.

17. There are some grammatical errors in the article, such as Line 32 and line 37, please check and revise. Please ask a native speaker for corrections in these cases.

18. The pictures in the text need to be further beautified.

19. I think the meaning of the research should be stated at the beginning of the abstract.

20. The proposed method looks like a combination of existing methods. What is the core innovation of this proposed method? Please explain this clear in the part of Introduction.

Author Response

(The authors gave the same response as above.)

Reviewer 3 Report

I have the following questions:

1. It is recommended to give a flowchart of the entire algorithm.

2. In addition to correcting the range variant motion error, this paper also corrects the remaining RCM, so how is this residual RCM introduced? Is it the range variant envelope error introduced by the range variant motion error?

3. In the case of squint mode or ultra-high resolution, RMA is required to complete the RCMC, but additional NsRCM is also introduced. It is recommended to explain it in the paper, which can refer to:

Wide-beam SAR autofocus based on blind RS. Sci China Inf Sci, 2022, doi: 10.1007/s11432-022-3574-7

4. In addition, in large scene conditions, the wide beam autofocus problem cannot be ignored, please explain.

Author Response

(The authors gave the same response as above.)

Reviewer 4 Report

The ms is well-written and clear and the topic is of great relevance. In my opinion, the ms can be recommended for publication once few major points are adequately addressed by the Authors.

- the state-of-the-art should be improved by critically discussing pro&co of the referenced methods. As it stands now it is a list of papers.

-Please, provide more info about the computational burden when solving (7).

There are effective iterative methods to solve (7) even in L_q space,

see e.g.; 0.1109/TGRS.2019.2908560, 10.1109/TGRS.2015.2458014

-Some symbols are not correctly displayed (e.g.; rows 118-119)

-If I understood correctly, the minimization is carried on in the Hilbert space; while sparsity is induced using a Tikhonov-like regularization term which is in the L1 space. I think the Author should explicitly refer that a fixed q=1  space is considered. For instance this is not clear in eq7 and in eq29. It seems that a variable-p approach is implemented.

- Is Fig5(c) correct? I understood that the method fails in reconstructing the exact location of the point targets but this panel is blank. Is this intentionally? Please clarify.

-How selecting lambda and mu.

-How the different methods behave when dealing with less sparse cases. For instance, what happens when reconstructing the background in Fig8?

Author Response

(The authors gave the same response as above.)

Round 2

Reviewer 2 Report

I have no questions. The paper has been improved a lot after the revision, it is recommended to accept.

Reviewer 3 Report

No more comments.